# Regularization Method for Singularly Perturbed Integro-Differential Equations with Rapidly Oscillating Coefficients and Rapidly Changing Kernels

**Burkhan Kalimbetov [1,*] and Valeriy Safonov [2]**

[1] Department of Mathematics, Akhmed Yassawi University, B. Sattarkhanov 29, Turkestan 161200, Kazakhstan
[2] Department of Higher Mathematics, National Research University «MPEI», Krasnokazarmennaya 14, 111250 Moscow, Russia; Singsaf@yandex.ru
[*] Correspondence: burkhan.kalimbetov@ayu.edu.kz

**Abstract:** In this paper, we consider a system with rapidly oscillating coefficients, which includes an integral operator with an exponentially varying kernel. The main goal of the work is to develop an algorithm for the regularization method for such systems and to identify the influence of the integral term on the asymptotic behavior of the solution of the original problem.

**Keywords:** singular perturbation; integro-differential equation; rapidly oscillating coefficient; regularization; asymptotic convergence; resonant exhibitors

**MSC:** 34K26; 45J05

## 1. Introduction

In the study of various issues related to dynamic stability, with the properties of media with a periodic structure, in the study of other applied problems, one has to deal with differential equations with rapidly oscillating coefficients. Equations of this kind can describe some mechanical or electrical systems that are under the influence of high-frequency external forces, automatic control systems with a linear adjustable object, etc. As an example, we can cite the principle of operation of an oscillator with a small mass and a nonlinear restoring force, in which a high-frequency periodic force with a large amplitude acts. The presence of high-frequency terms creates serious problems for their direct numerical solutions. Therefore, asymptotic methods are usually applied to such equations first, the most famous of which are the Feshchenko–Shkil–Nikolenko splitting method [1–5] and the Lomov's regularization method [6–8]. It should also be noted that singularly perturbed equations are the object of study by several Russian researchers, as well as other scientists (see, for example [9–22]).

In this paper, the Lomov's regularization method is generalized to previously unexplored integro-differential equations with rapidly oscillating coefficients and with rapidly decreasing kernels of the form

$$\varepsilon\frac{dz}{dt} - a(t)z - \varepsilon g(t)\cos\frac{\beta(t)}{\varepsilon}z - \int_{t_0}^t e^{\frac{1}{\varepsilon}\int_s^t \mu(\theta)d\theta}K(t,s)z(s,\varepsilon)ds = h(t),\ z(t_0,\varepsilon) = z^0,\ t \in [t_0,T] \quad (1)$$

where $z = z(t,\varepsilon)$, $h(t)$, $\beta'(t) > 0$, $a(t) > 0$, $\mu(t) < 0$, $a(t) \neq \mu(t)\ (\forall t \in [t_0,T])$, $g(t)$ are scalar functions, $z^0$ is a constant, $\varepsilon > 0$ is a small parameter. In the case $\beta(t) = 2\gamma(t)$, and of the absence of an integral term, such a system was considered in [6–8].

The limit operator $a(t)$ has a spectrum $\lambda_1(t) = a(t)$, functions $\lambda_2(t) = -i\beta'(t)$ and $\lambda_3(t) = +i\beta'(t)$ are associated with the presence in Equation (1) of a rapidly oscillating $\cos\frac{\beta(t)}{\varepsilon}$, and the function $\lambda_4(t) = \mu(t)$ characterizes the rapid change in the kernel of the integral operator.

We introduce the following notations:

$\lambda(t) = (\lambda_1(t), ..., \lambda_4(t))$,

$m = (m_1, ..., m_4)$ is multi-index with non-negative components $m_j$, $j = \overline{1,4}$,

$|m| = \sum_{j=1}^{4} m_j$ is multi-index height $m$,

$(m, \lambda(t)) = \sum_{j=1}^{4} m_j \lambda_j(t)$.

Assume that the following conditions are met:

(1) $a(t), \beta(t), \mu(t) \in C^\infty([t_0, T], \mathbb{R})$, $g(t), h(t) \in C^\infty([t_0, T], \mathbb{C})$,
　　$K(t,s) \in C^\infty\{t_0 \le s \le t \le T, \mathbb{C}\}$;

(2) the relations $(m, \lambda(t)) = 0$, $(m, \lambda(t)) = \lambda_j(t)$, $j \in \{1, ..., 4\}$ for all multi-indices $m$ with $|m| \ge 2$ or are not fulfilled for any $t \in [t_0, T]$, or are fulfilled identically on the whole segment $t \in [t_0, T]$.

In other words, resonant multi-indices are exhausted by the following sets

$$\Gamma_0 = \{m : (m, \lambda(t)) \equiv 0, |m| \ge 2, \forall t \in [t_0, T]\},$$
$$\Gamma_j = \{m : (m, \lambda(t)) \equiv \lambda_j(t), |m| \ge 2, \forall t \in [t_0, T]\}, \ j = \overline{1,4}.$$

Under these conditions, we will develop an algorithm for constructing a regularized [6] asymptotic solution of the problem (1).

## 2. Regularization of the Problem (1)

Denote by $\sigma_j = \sigma_j(\varepsilon)$ independent of the $t$ quantities $\sigma_1 = e^{-\frac{i}{\varepsilon}\beta(t_0)}$, $\sigma_2 = e^{+\frac{i}{\varepsilon}\beta(t_0)}$, and rewrite the Equation (1) in the form

$$
\mathbf{L}\, z(t, \varepsilon) \equiv \varepsilon \frac{dz}{dt} - a(t)z - \varepsilon \frac{g(t)}{2}\left( e^{-\frac{i}{\varepsilon}\int_{t_0}^{t}\beta'(\theta)d\theta}\sigma_1 + e^{+\frac{i}{\varepsilon}\int_{t_0}^{t}\beta'(\theta)d\theta}\sigma_2 \right) z -
$$
$$
- \int_{t_0}^{t} e^{\frac{1}{\varepsilon}\int_{s}^{t}\mu(\theta)d\theta}K(t,s)z(s,\varepsilon)ds = h(t), \quad z(t_0, \varepsilon) = z^0, \quad t \in [t_0, T]. \tag{2}
$$

We introduce regularizing variables

$$\tau_j = \frac{1}{\varepsilon}\int_{t_0}^{t}\lambda_j(\theta)d\theta \equiv \frac{\psi_j(t)}{\varepsilon}, \ j = \overline{1,4} \tag{3}$$

and instead of problem (2) we consider the problem

$$
\mathbf{L}\, \tilde{z}(t, \tau, \varepsilon) \equiv \varepsilon \frac{\partial \tilde{z}}{\partial t} + \sum_{j=1}^{4}\lambda_j(t)\frac{\partial \tilde{z}}{\partial \tau_j} - \lambda_1(t)\tilde{z} - \varepsilon \frac{g(t)}{2}\left( e^{\tau_2}\sigma_1 + e^{\tau_3}\sigma_2 \right)\tilde{z} -
$$
$$
- \int_{t_0}^{t} e^{\frac{1}{\varepsilon}\int_{s}^{t}\lambda_4(\theta)d\theta}K(t,s)\tilde{z}\left( s, \frac{\psi(s)}{\varepsilon}, \varepsilon \right)ds = h(t), \ \tilde{z}(t, \tau, \varepsilon)|_{t=t_0, \tau=0} = z^0, \quad t \in [t_0, T] \tag{4}
$$

for the function $\tilde{z} = \tilde{z}(t, \tau, \varepsilon)$, where it is indicated (according to (3)): $\tau = (\tau_1, ..., \tau_4)$, $\psi = (\psi_1, ..., \psi_4)$. It is clear that if $\tilde{z} = \tilde{z}(t, \tau, \varepsilon)$ is the solution of the problem (4), then the function $z = \tilde{z}\left( t, \frac{\psi(t)}{\varepsilon}, \varepsilon \right)$ is an exact solution of the problem (2), therefore, the problem (4) is an extension of the problem (2).

However, (4) cannot be considered completely regularized, since the integral term

$$J\tilde{z} = \int_{t_0}^{t} e^{\frac{1}{\varepsilon}\int_{s}^{t}\lambda_4(\theta)d\theta}K(t,s)\,\tilde{z}\left( s, \frac{\psi(s)}{\varepsilon}, \varepsilon \right)ds$$

has not been regularized in it. To regularize $J$, we introduce a class $M_\varepsilon$, asymptotically invariant with respect to the operator $J\tilde{z}$ (see [6]; p. 62).

We first consider the space $U$ of functions $z(t, \tau)$, representable by sums

$$
\begin{aligned}
z(t, \tau, \sigma) &= z_0(t, \sigma) + \sum_{i=1}^4 z_i(t, \sigma) e^{\tau_i} + \sum_{2 \leq |m| \leq N_z}^* z^m(t, \sigma) e^{(m, \tau)}, \\
z_0(t, \sigma), z_i(t, \sigma), z^m(t, \sigma) &\in C^\infty([t_0, T], \mathbb{C}), \quad i = \overline{1, 4}, \ 2 \leq |m| \leq N_z
\end{aligned}
\tag{5}
$$

where the asterisk $*$ above the sum sign indicates that in it the summation for $|m| \geq 2$ occurs only over nonresonant multi-indices $m = (m_1, ..., m_4)$, i.e., over $m \notin \bigcup_{i=0}^4 \Gamma_i$.

Note that in (5) the degree $N_z$ of the polynomial $z(t, \tau, \sigma)$ to exponentials $e^{\tau_j}$ depends on the element $z$. The elements of the space $U$ depend on bounded in $\varepsilon > 0$ constants $\sigma_1 = \sigma_1(\varepsilon)$ and $\sigma_2 = \sigma_2(\varepsilon)$, which do not affect the development of the algorithm described below, therefore in the notation of element (5) of this space $U$ we omit the dependence on $\sigma = (\sigma_1, \sigma_2)$ for brevity. We show that the class $M_\varepsilon = U|_{\tau = \psi(t)/\varepsilon}$ is asymptotically invariant with respect to the operator $J$.

The image of the operator $J$ on the element (5) of the space $U$ has the form:

$$
Jz(t, \tau) = \int_{t_0}^t e^{\frac{1}{\varepsilon} \int_s^t \lambda_4(\theta) d\theta} K(t, s) z_0(s) \, ds + \sum_{i=1}^4 \int_{t_0}^t e^{\frac{1}{\varepsilon} \int_s^t \lambda_4(\theta) d\theta} K(t, s) z_i(s) e^{\frac{1}{\varepsilon} \int_{t_0}^s \lambda_i(\theta) d\theta} \, ds +
$$

$$
+ \sum_{2 \leq |m| \leq N_z}^* \int_{t_0}^t e^{\frac{1}{\varepsilon} \int_s^t \lambda_4(\theta) d\theta} K(t, s) z^m(s) e^{\frac{1}{\varepsilon} \int_{t_0}^s (m, \lambda(\theta)) d\theta} \, ds =
$$

$$
= \int_{t_0}^t e^{\frac{1}{\varepsilon} \int_s^t \lambda_4(\theta) d\theta} K(t, s) z_0(s) \, ds + e^{\frac{1}{\varepsilon} \int_{t_0}^t \lambda_4(\theta) d\theta} \int_{t_0}^t K(t, s) z_4(s) \, ds +
$$

$$
+ \sum_{i=1, i \neq 4}^4 e^{\frac{1}{\varepsilon} \int_{t_0}^t \lambda_4(\theta) d\theta} \int_{t_0}^t K(t, s) z_i(s) e^{\frac{1}{\varepsilon} \int_{t_0}^s (\lambda_i(\theta) - \lambda_4(\theta)) d\theta} \, ds +
$$

$$
+ \sum_{2 \leq |m| \leq N_z}^* e^{\frac{1}{\varepsilon} \int_{t_0}^t \lambda_4(\theta) d\theta} \int_{t_0}^t K(t, s) z^m(s) e^{\frac{1}{\varepsilon} \int_{t_0}^s (m - e_4, \lambda(\theta)) d\theta} \, ds.
$$

Integrating in parts, we have

$$
J_0(t, \varepsilon) = \int_{t_0}^t K(t, s) z_0(s) e^{\frac{1}{\varepsilon} \int_{t_0}^s \lambda_4(\theta) d\theta} \, ds = \varepsilon \int_{t_0}^t \frac{K(t, s) z_0(s)}{\lambda_4(s)} de^{\frac{1}{\varepsilon} \int_{t_0}^s \lambda_4(\theta) d\theta} =
$$

$$
= \varepsilon \frac{K(t, s) z_0(s)}{\lambda_4(s)} e^{\frac{1}{\varepsilon} \int_{t_0}^s \lambda_4(\theta) d\theta} \Big|_{s=t_0}^{s=t} - \varepsilon \int_{t_0}^t \left( \frac{\partial}{\partial s} \frac{K(t, s) z_0(s)}{\lambda_4(s)} \right) e^{\frac{1}{\varepsilon} \int_{t_0}^s \lambda_4(\theta) d\theta} \, ds =
$$

$$
= \varepsilon \left[ \frac{K(t, t) z_0(t)}{\lambda_4(t)} e^{\frac{1}{\varepsilon} \int_{t_0}^t \lambda_4(\theta) d\theta} - \frac{K(t, t_0) z_0(t_0)}{\lambda_4(t_0)} \right] - \varepsilon \int_{t_0}^t \left( \frac{\partial}{\partial s} \frac{K(t, s) z_0(s)}{\lambda_4(s)} \right) e^{\frac{1}{\varepsilon} \int_{t_0}^s \lambda_4(\theta) d\theta} \, ds.
$$

Continuing this process further, we obtained the decomposition

$$
J_0(t, \varepsilon) = \sum_{\nu=0}^\infty (-1)^\nu \varepsilon^{\nu+1} \left[ \left( I_0^\nu (K(t, s) z_0(s)) \right)_{s=t} e^{\frac{1}{\varepsilon} \int_{t_0}^t \lambda_4(\theta) d\theta} - \left( I_0^\nu (K(t, s) z_0(s)) \right)_{s=t_0} \right],
$$

$$
\boxed{I_0^0 = \frac{1}{\lambda_4(s)} \cdot, \quad I_0^\nu = \frac{1}{\lambda_4(s)} \frac{\partial}{\partial s} I_0^{\nu-1} \ (\nu \geq 1)}.
$$

Next, apply the same operation to the integrals:

$$
J_{4,i}(t, \varepsilon) = e^{\frac{1}{\varepsilon} \int_{t_0}^t \lambda_4(\theta) d\theta} \int_{t_0}^t K(t, s) z_i(s) e^{\frac{1}{\varepsilon} \int_{t_0}^s (\lambda_i(\theta) - \lambda_4(\theta)) d\theta} \, ds =
$$

$$
= \varepsilon e^{\frac{1}{\varepsilon} \int_{t_0}^t \lambda_4(\theta) d\theta} \int_{t_0}^t \frac{K(t, s) z_i(s)}{\lambda_i(s) - \lambda_4(s)} de^{\frac{1}{\varepsilon} \int_{t_0}^s (\lambda_i(\theta) - \lambda_4(\theta)) d\theta} =
$$

$$
= \varepsilon e^{\frac{1}{\varepsilon} \int_{t_0}^{t} \lambda_4(\theta) d\theta} \left[ \left. \frac{K(t,s) z_i(s)}{\lambda_i(s) - \lambda_4(s)} e^{\frac{1}{\varepsilon} \int_{t_0}^{s} (\lambda_i(\theta) - \lambda_4(\theta)) d\theta} \right|_{s=t_0}^{s=t} - \right.
$$

$$
\left. - \varepsilon \int_{t_0}^{t} \left( \frac{\partial}{\partial s} \frac{K(t,s) z_i(s)}{\lambda_i(s) - \lambda_4(s)} \right) e^{\frac{1}{\varepsilon} \int_{t_0}^{s} (\lambda_i(\theta) - \lambda_4(\theta)) d\theta} ds \right] =
$$

$$
= \varepsilon \left[ \frac{K(t,t) z_i(t)}{\lambda_i(t) - \lambda_4(t)} e^{\frac{1}{\varepsilon} \int_{t_0}^{t} \lambda_i(\theta) d\theta} - \frac{K(t,t_0) z_i(t_0)}{\lambda_i(t_0) - \lambda_4(t_0)} e^{\frac{1}{\varepsilon} \int_{t_0}^{t} \lambda_4(\theta) d\theta} \right] -
$$

$$
- \varepsilon e^{\frac{1}{\varepsilon} \int_{t_0}^{t} \lambda_4(\theta) d\theta} \int_{t_0}^{t} \left( \frac{\partial}{\partial s} \frac{K(t,s) z_i(s)}{\lambda_i(s) - \lambda_4(s)} \right) e^{\frac{1}{\varepsilon} \int_{t_0}^{s} (\lambda_i(\theta) - \lambda_4(\theta)) d\theta} ds =
$$

$$
= \sum_{\nu=0}^{\infty} (-1)^{\nu} \varepsilon^{\nu+1} \left[ \left( I_i^{\nu} (K(t,s) z_i(s)) \right)_{s=t} e^{\frac{1}{\varepsilon} \int_{t_0}^{t} \lambda_i(\theta) d\theta} - \left( I_i^{\nu} (K(t,s) z_i(s)) \right)_{s=t_0} e^{\frac{1}{\varepsilon} \int_{t_0}^{t} \lambda_4(\theta) d\theta} \right],
$$

$$
\boxed{I_i^0 = \frac{1}{\lambda_i(s) - \lambda_4(s)} , \quad I_i^{\nu} = \frac{1}{\lambda_i(s) - \lambda_4(s)} \frac{\partial}{\partial s} I_i^{\nu-1}, \ \nu \geq 1, \ i = \overline{1,3}.}
$$

Denote bay $e_4 = (0,0,0,1)$. Then

$$
J_m(t,\varepsilon) = e^{\frac{1}{\varepsilon} \int_{t_0}^{t} \lambda_4(\theta) d\theta} \int_{t_0}^{t} K(t,s) z^m(s) e^{\frac{1}{\varepsilon} \int_{t_0}^{s} (m - e_4, \lambda(\theta)) d\theta} ds =
$$

$$
= \varepsilon e^{\frac{1}{\varepsilon} \int_{t_0}^{t} \lambda_4(\theta) d\theta} \int_{t_0}^{t} \frac{K(t,s) z^m(s)}{(m - e_4, \lambda(s))} d e^{\frac{1}{\varepsilon} \int_{t_0}^{s} (m - e_4, \lambda(\theta)) d\theta} =
$$

$$
= \varepsilon e^{\frac{1}{\varepsilon} \int_{t_0}^{t} \lambda_4(\theta) d\theta} \left[ \left. \frac{K(t,s) z^m(s)}{(m - e_4, \lambda(s))} e^{\frac{1}{\varepsilon} \int_{t_0}^{s} (m - e_4, \lambda(\theta)) d\theta} \right|_{s=t_0}^{s=t} - \right.
$$

$$
\left. - \int_{t_0}^{t} \left( \frac{\partial}{\partial s} \frac{K(t,s) z^m(s)}{(m - e_4, \lambda(s))} \right) e^{\frac{1}{\varepsilon} \int_{t_0}^{s} (m - e_4, \lambda(\theta)) d\theta} ds \right] =
$$

$$
= \sum_{\nu=0}^{\infty} (-1)^{\nu} \varepsilon^{\nu+1} \left[ \left( I_{4,m}^{\nu} (K(t,s) z^m(s)) \right)_{s=t} e^{\frac{1}{\varepsilon} \int_{t_0}^{t} (m, \lambda(\theta)) d\theta} - \right.
$$

$$
\left. - \left( I_{4,m}^{\nu} (K(t,s) z^m(s)) \right)_{s=t_0} e^{\frac{1}{\varepsilon} \int_{t_0}^{t} \lambda_4(\theta) d\theta} \right],
$$

$$
\boxed{\begin{array}{l} I_{4,m}^0 = \dfrac{1}{(m - e_4, \lambda(s))} , \ I_{4,m}^{\nu} = \dfrac{1}{(m - e_4, \lambda(s))} \dfrac{\partial}{\partial s} I_{4,m}^{\nu-1}, \nu \geq 1, \\[2mm] 2 \leq |m| \leq N_z. \end{array}}
$$

Here it is taken into account that $(m - e_4, \lambda(s)) \neq 0$, since by the definition of the space $U$ multi-indices $m \notin \Gamma_4$. The image of the operator $J$ on the space $U$ element (5) is represented as a series

$$
Jz(t,\tau) = e^{\frac{1}{\varepsilon} \int_{t_0}^{t} \lambda_4(\theta) d\theta} \int_{t_0}^{t} K(t,s) z_4(s) ds + \sum_{\nu=0}^{\infty} (-1)^{\nu} \varepsilon^{\nu+1} \left[ \left( I_0^{\nu} (K(t,s) z_0(s)) \right)_{s=t} e^{\frac{1}{\varepsilon} \int_{t_0}^{t} \lambda_4(\theta) d\theta} - \right.
$$

$$
\left. - \left( I_0^{\nu} (K(t,s) z_0(s)) \right)_{s=t_0} \right] + \sum_{i=1, i \neq 4}^{4} \sum_{\nu=0}^{\infty} (-1)^{\nu} \varepsilon^{\nu+1} \left[ \left( I_i^{\nu} (K(t,s) z_i(s)) \right)_{s=t} e^{\frac{1}{\varepsilon} \int_{t_0}^{t} \lambda_i(\theta) d\theta} - \right.
$$

$$
\left. - \left( I_i^{\nu} (K(t,s) z_i(s)) \right)_{s=t_0} e^{\frac{1}{\varepsilon} \int_{t_0}^{t} \lambda_4(\theta) d\theta} \right] +
$$

$$
+ \sum_{2 \leq |m| \leq N_z}^{*} \sum_{\nu=0}^{\infty} (-1)^{\nu} \varepsilon^{\nu+1} \left[ \left( I_{4,m}^{\nu} (K(t,s) z^m(s)) \right)_{s=t} e^{\frac{1}{\varepsilon} \int_{t_0}^{t} (m, \lambda(\theta)) d\theta} - \right.
$$

$$
\left. - \left( I_{4,m}^{\nu} (K(t,s) z^m(s)) \right)_{s=t_0} e^{\frac{1}{\varepsilon} \int_{t_0}^{t} \lambda_4(\theta) d\theta} \right], \tau = \psi(t) / \varepsilon.
$$

It is easy to show (see, for example, [23], pp. 291–294) that this series converges asymptotically for $\varepsilon \to +0$ (uniformly in $t \in [t_0, T]$). This means that the class $M_\varepsilon$ is asymptotically invariant (for $\varepsilon \to +0$) with respect to the operator $J$.

Let as introduce the operators $R_\nu : U \to U$, acting on each element $z(t, \tau) \in U$ of the form (5) according to the law:

$$R_0 z(t, \tau) = e^{\tau_4} \int_{t_0}^{t} K(t, s) z_4(s) \, ds, \tag{$6_0$}$$

$$R_1 z(t, \tau) = \left[ \left( I_0^0 (K(t, s) z_0(s)) \right)_{s=t} e^{\tau_4} - \left( I_0^0 (K(t, s) z_0(s)) \right)_{s=t_0} \right] +$$

$$+ \sum_{i=1}^{3} \left[ \left( I_i^0 (K(t, s) z_i(s)) \right)_{s=t} e^{\tau_i} - \left( I_i^0 (K(t, s) z_i(s)) \right)_{s=t_0} e^{\tau_4} \right] + \tag{$6_1$}$$

$$+ \sum_{2 \leq |m| \leq N_z}^{*} \left[ \left( I_{4,m}^0 (K(t, s) z^m(s)) \right)_{s=t} e^{(m, \tau)} - \left( I_{4,m}^0 (K(t, s) z^m(s)) \right)_{s=t_0} e^{\tau_4} \right],$$

$$R_{\nu+1} z(t, \tau) = \left[ \left( I_0^\nu (K(t, s) z_0(s)) \right)_{s=t} e^{\tau_4} - \left( I_0^\nu (K(t, s) z_0(s)) \right)_{s=t_0} \right] +$$

$$+ \sum_{i=1}^{3} (-1)^\nu \left[ \left( I_i^\nu (K(t, s) z_i(s)) \right)_{s=t} e^{\tau_i} - \left( I_i^\nu (K(t, s) z_i(s)) \right)_{s=t_0} e^{\tau_4} \right] + \tag{$6_{\nu+1}$}$$

$$+ \sum_{2 \leq |m| \leq N_z}^{*} \left[ \left( I_{4,m}^\nu (K(t, s) z^m(s)) \right)_{s=t} e^{(m, \tau)} - \left( I_{4,m}^\nu (K(t, s) z^m(s)) \right)_{s=t_0} e^{\tau_4} \right], \nu \geq 1.$$

Let now $\tilde{z}(t, \tau, \varepsilon)$ be an arbitrary continuous function in $(t, \tau) \in [t_0, T] \times \{\tau : Re\tau_j \leq 0, j = \overline{1,4}\}$ with the asymptotic expansion

$$\tilde{z}(t, \tau, \varepsilon) = \sum_{k=0}^{\infty} \varepsilon^k z_k(t, \tau), \quad z_k(t, \tau) \in U, \tag{7}$$

converging as $\varepsilon \to +0$ (uniformly in $(t, \tau) \in [t_0, T] \times \{\tau : Re\tau_j \leq 0, j = \overline{1,4}\}$). Then the image $J\tilde{z}(t, \tau, \varepsilon)$ of this function is expanded in the asymptotic series

$$J\tilde{z}(t, \tau, \varepsilon) = \sum_{k=0}^{\infty} \varepsilon^k Jz_k(t, \tau) = \sum_{r=0}^{\infty} \varepsilon^r \sum_{s=0}^{r} R_{r-s} z_s(t, \tau) \big|_{\tau = \psi(t)/\varepsilon}.$$

This equality is the basis for introducing the extension of the operator $J$ on the series type (7):

$$\tilde{J}\tilde{z}(t, \tau, \varepsilon) \equiv \tilde{J} \left( \sum_{k=0}^{\infty} \varepsilon^k z_k(t, \tau) \right) \stackrel{def}{=} \sum_{r=0}^{\infty} \varepsilon^r \sum_{s=0}^{r} R_{r-s} z_s(t, \tau).$$

Although the operator $\tilde{J}$ is formally defined, its usefulness is obvious, since in practice they usually construct the $N$-th approximation of the asymptotic solution of problem (2), in which only the $N$-th partial sums of the series (7) will take part, which do not have a formal but true meaning. Now we can write down a problem that is completely regularized with respect to the original problem (2):

$$\mathbf{L}\tilde{z}(t, \tau, \varepsilon) \equiv \varepsilon \frac{\partial \tilde{z}}{\partial t} + \sum_{j=1}^{4} \lambda_j(t) \frac{\partial \tilde{z}}{\partial \tau_j} - \lambda_1(t)\tilde{z} - \varepsilon \frac{g(t)}{2} \left( e^{\tau_2} \sigma_1 + e^{\tau_3} \sigma_2 \right) \tilde{z} - \tilde{J}\tilde{z} = h(t),$$

$$\tilde{z}(t, \tau, \varepsilon)|_{t=t_0, \tau=0} = z^0, \quad t \in [t_0, T]. \tag{8}$$

### 3. Iterative Problems and Their Solvability in the Space $U$

Substituting series (7) into (8) and equating the coefficients for the same powers $\varepsilon$, we obtain the following iterative problems:

$$\mathbf{L}\, z_0\,(t,\tau) \equiv \sum_{j=1}^{4} \lambda_j\,(t)\,\frac{\partial z_0}{\partial \tau_j} - \lambda_1(t)z_0 - R_0 z_0 = h\,(t)\,,\ z_0\,(t_0,0) = z^0; \tag{$9_0$}$$

$$\mathbf{L}\, z_1\,(t,\tau) = -\frac{\partial z_0}{\partial t} + \frac{g(t)}{2}\,\left(e^{\tau_2}\sigma_1 + e^{\tau_3}\sigma_2\right)z_0 + R_1 z_0,\ z_1\,(t_0,0) = 0; \tag{$9_1$}$$

$$\mathbf{L}\, z_2\,(t,\tau) = -\frac{\partial z_1}{\partial t} + \frac{g(t)}{2}\,\left(e^{\tau_2}\sigma_1 + e^{\tau_3}\sigma_2\right)z_1 + R_1 z_1 + R_2 z_0,\ z_0\,(t_0,0) = 0; \tag{$9_2$}$$

$$\cdots$$

$$\mathbf{L}\, z_k\,(t,\tau) = -\frac{\partial z_{k-1}}{\partial t} + \frac{g(t)}{2}\,\left(e^{\tau_2}\sigma_1 + e^{\tau_3}\sigma_2\right)z_{k-1} + R_k z_0 + ... + R_1 z_{k-1},\ z_k\,(t_0,0) = 0,\ k \geq 1. \tag{$9_k$}$$

Each of the iterative problems can be written as

$$\mathbf{L}\, z\,(t,\tau) \equiv \sum_{j=1}^{4} \lambda_j\,(t)\,\frac{\partial z}{\partial \tau_j} - \lambda_1(t)z - R_0 z = H\,(t,\tau)\,,\ z\,(t_0,0) = z^*, \tag{10}$$

where $H\,(t,\tau) = H_0\,(t) + \sum_{i=1}^{4} H_i\,(t)\,e^{\tau_i} + \sum_{2\leq|m|\leq N_H}^{*} H^m\,(t)\,e^{(m,\tau)}$ is the known function of the space $U$, $z^*$ is the known number of complex the space $\mathbb{C}$, and the operator $R_0$ has the form (see $(6_0)$)

$$R_0 z \equiv R_0\,\left(z_0\,(t) + \sum_{i=1}^{4} z_i\,(t)\,e^{\tau_i} + \sum_{2\leq|m|\leq N_z}^{*} z^m\,(t)\,e^{(m,\tau)}\right) \overset{def}{=} e^{\tau_4} \int_{t_0}^{t} K\,(t,s)\,z_4\,(s)\,ds.$$

We introduce the scalar product (for each $t \in [t_0, T]$) in the space $U$:

$$< z, w > \equiv\, < z_0\,(t) + \sum_{i=1}^{4} z_i\,(t)\,e^{\tau_i} + \sum_{2\leq|m|\leq N_z}^{*} z^m\,(t)\,e^{(m,\tau)},$$

$$w_0\,(t) + \sum_{i=1}^{4} w_i\,(t)\,e^{\tau_i} + \sum_{2\leq|m|\leq N_w}^{*} w^m\,(t)\,e^{(m,\tau)} > \overset{def}{=}$$

$$\overset{def}{=} (z_0\,(t)\,,\, w_0\,(t)) + \sum_{i=1}^{4} (z_i\,(t)\,,\, w_i\,(t)) + \sum_{2\leq|m|\leq \min(N_z,N_w)}^{*} (z^m\,(t)\,,\, w^m\,(t))\,,$$

where $(*, *)$ we denote the ordinary scalar product in the complex space $\mathbb{C}$: $(u, v) = u \cdot \bar{v}$. We prove the following statement.

**Theorem 1.** *Suppose that conditions (1) and (2) are satisfied and the right-hand side $H\,(t,\tau) = H_0\,(t) + \sum_{i=1}^{4} H_i\,(t)\,e^{\tau_i} + \sum_{2\leq|m|\leq N_H}^{*} H^m\,(t)\,e^{(m,\tau)}$ of the Equation (10) belongs to the space $U$. Then for the solvability of the Equation (10) in $U$ it is necessary and sufficient that the identities*

$$< H\,(t,\tau)\,,\, e^{\tau_1} > \equiv 0,\quad \forall t \in [t_0, T] \tag{11}$$

*hold true.*

**Proof.** We will determine the solution of the Equation (10) in the form of an element (5) of the space $U$:

$$z(t,\tau) = z_0(t) + \sum_{i=1}^{4} z_i(t) e^{\tau_i} + \sum_{2 \le |m| \le N_H}^{*} z^m(t) e^{(m,\tau)}. \tag{12}$$

Substituting (12) into the Equation (10), we have

$$-\lambda_1(t) z_0(t) + \sum_{i=1}^{4} [\lambda_i(t) - \lambda_1(t)] z_i(t) e^{\tau_i} + \sum_{2 \le |m| \le N_H}^{*} [(m,\lambda(t)) - \lambda_1(t)] z^m(t) e^{(m,\tau)} -$$

$$-e^{\tau_4} \int_{t_0}^{t} K(t,s) z_4(s) ds = H_0(t) + \sum_{i=1}^{4} H_i(t) e^{\tau_i} + \sum_{2 \le |m| \le N_H}^{*} H^m(t) e^{(m,\tau)}.$$

Equating here separately the free terms and coefficients at the same exponents, we obtained the following equations:

$$-\lambda_1(t) z_0(t) = H_0(t), \tag{$13_0$}$$

$$[\lambda_i(t) - \lambda_1(t)] z_i(t) = H_i(t), \ i = \overline{1,3}; \tag{$13_i$}$$

$$[\lambda_4(t) - \lambda_1(t)] z_4(t) - \int_{t_0}^{t} K(t,s) z_4(s) ds = H_4(t); \tag{$13_4$}$$

$$[(m,\lambda(t)) - \lambda_1(t)] z^m(t) = H^m(t), \ m \notin \Gamma_1, \ 2 \le |m| \le N_H. \tag{$13_m$}$$

Since the function $\lambda_1(t) \ne 0 \ \forall t \in [t_0, T]$, the Equation ($13_0$) has a unique solution $z_0(t) = -\lambda_1^{-1}(t) H_0(t)$. Since $\lambda_4(t) - \lambda_1(t) \ne 0 \ \forall t \in [t_0, T]$, then the Equation ($13_4$) can be written as

$$z_4(t) = \int_{t_0}^{t} \left( [\lambda_4(t) - \lambda_1(t)]^{-1} K(t,s) \right) z_4(s) ds - [\lambda_4(t) - \lambda_1(t)]^{-1} H_4(t). \tag{14}$$

Due to the smoothness of the kernel $\left( [\lambda_4(t) - \lambda_1(t)]^{-1} K(t,s) \right)$ and heterogeneity $-[\lambda_4(t) - \lambda_1(t)]^{-1} H_4(t)$, this Volterra integral equation has a unique solution $z_4(t) \in C^\infty([t_0, T], \mathbb{C})$. The Equations ($13_2$) and ($13_3$) also have unique solutions

$$z_i(t) = [\lambda_i(t) - \lambda_1(t)]^{-1} H_i(t) \in C^\infty([t_0, T], \mathbb{C}), \ i = 2,3,$$

since $\lambda_i(t) \ne \lambda_1(t), i = 2,3$. The Equation ($13_1$) is solvable in the space $C^\infty([t_0, T], \mathbb{C})$ if and only if identities $(H_1(t), e^{\tau_1}) \equiv 0 \ \forall t \in [t_0, T]$ hold. It is easy to see that this identity coincides with identity (11).

Further, since $(m,\lambda(t)) \ne \lambda_1(t), \ 2 \le |m| \le N_H \ (\forall m \notin \Gamma_1)$, then the Equation ($13_m$) has a unique solution

$$z^m(t) = [(m,\lambda(t)) I - A(t)]^{-1} H^m(t) \in C^\infty([t_0, T], \mathbb{C}), \ 2 \le |m| \le N_H.$$

Thus, condition (11) is necessary and sufficient for the solvability of the Equation (10) in the space $U$. The Theorem 1 is proved. $\square$

**Remark 1.** *If identity (11) holds, then under conditions (1) and (2), the Equation (10) has the following solution in the space $U$ :*

$$z(t,\tau) = z_0(t) + \sum_{i=1}^{4} z_i(t) e^{\tau_i} + \sum_{2 \le |m| \le N_H}^{*} z^m(t) e^{(m,\tau)} \equiv z_0(t) + \alpha_1(t) e^{\tau_1} +$$
$$+ h_{21}(t) e^{\tau_2} + h_{31}(t) e^{\tau_3} + z_4(t) e^{\tau_4} + \sum_{2 \le |m| \le N_H}^{*} P^m(t) e^{(m,\tau)}, \tag{15}$$

where $\alpha_1(t) \in C^\infty([t_0, T], \mathbb{C})$ *are arbitrary function*, $z_0(t) = -\lambda_1^{-1}(t)H_0(t)$, $z_4(t)$ *is the solution of the integral Equation* (14), *and introduced notations*

$$h_{21}(t) \equiv \frac{H_2(t)}{\lambda_2(t) - \lambda_1(t)}, \quad h_{31}(t) \equiv \frac{H_3(t)}{\lambda_3(t) - \lambda_1(t)}, \quad P^m(t) \equiv [(m, \lambda(t)) - \lambda_1(t)]^{-1} H^m(t).$$

## 4. The Remainder Term Theorem

Along with problem (10), we consider the equation

$$\mathbf{L}\, w(t, \tau) = -\frac{\partial z}{\partial t} + \frac{g(t)}{2}(e^{\tau_2}\sigma_1 + e^{\tau_3}\sigma_2)z + R_1 z + Q(t, \tau), \tag{16}$$

where $z = z(t, \tau)$ is the solution (15) of Equation (10), $Q(t, \tau) \in U$ is the known function of the space $U$ (this form will have problems $(9_{k+1})$ after calculating the solution of the problem $(9_k)$ in $U$). The right side of this equation:

$$G(t, \tau) \equiv -\frac{\partial z}{\partial t} + \frac{g(t)}{2}(e^{\tau_2}\sigma_1 + e^{\tau_3}\sigma_2)z + R_1 z + Q(t, \tau) =$$

$$= -\frac{\partial}{\partial t}\left[ z_0(t) + \sum_{i=1}^{4} z_i(t)e^{\tau_i} + \sum_{2 \leq |m| \leq N_H}^{*} z^m(t)e^{(m,\tau)} \right] +$$

$$+\frac{g(t)}{2}(e^{\tau_2}\sigma_1 + e^{\tau_3}\sigma_2)\left[ z_0(t) + \sum_{i=1}^{4} z_i(t)e^{\tau_i} + \sum_{2 \leq |m| \leq N_H}^{*} z^m(t)e^{(m,\tau)} \right] + R_1 z + Q(t, \tau),$$

may not belong to the space $U$, if $z = z(t, \tau) \in U$. Indeed, taking into account the form (15) of function $z = z(t, \tau) \in U$, we consider in $G(t, \tau)$, for example, the terms

$$Z(t, \tau) \equiv \frac{g(t)}{2}(e^{\tau_2}\sigma_1 + e^{\tau_3}\sigma_2)\left[ z_0(t) + \sum_{i=1}^{4} z_i(t)e^{\tau_i} + \sum_{2 \leq |m| \leq N_H}^{*} z^m(t)e^{(m,\tau)} \right] =$$

$$= \frac{g(t)}{2}z_0(t)(e^{\tau_2}\sigma_1 + e^{\tau_3}\sigma_2) + \sum_{i=1}^{4}\frac{g(t)}{2}z_i(t)\left(e^{\tau_i + \tau_2}\sigma_1 + e^{\tau_i + \tau_3}\sigma_2\right) +$$

$$+\frac{g(t)}{2}(e^{\tau_2}\sigma_1 + e^{\tau_3}\sigma_2)\sum_{2 \leq |m| \leq N_H}^{*} P^m(t)e^{(m,\tau)}.$$

Function $Z(t, \tau) \notin U$, since it contains resonant exponentials $e^{\tau_2 + \tau_3} = e^{(m,\tau)}|_{m=(0,1,1,0)}, e^{\tau_2 + (m,\tau)}(m_2 + 1 = m_3), e^{\tau_3 + (m,\tau)}(m_3 + 1 = m_2)$, and, therefore, the right-hand side $G(t, \tau) = Z(t, \tau)$ of the Equation (16) also does not belong to the $U$. Then, according to the well-known theory (see [6], p. 234), we need to embed $\wedge: G(t, \tau) \to \hat{G}(t, \tau)$ the right-hand side $G(t, \tau)$ of the Equation (16) into the space $U$. This operation is defined as follows.

Let the function $G(t, \tau) = \sum_{|m|=0}^{N} w^m(t)e^{(m,\tau)}$ contain resonant exponentials, i.e., $G(t, \tau)$, it has the form

$$G(t, \tau) = w_0(t) + \sum_{i=1}^{4} w_i(t)e^{\tau_i} + \sum_{j=0}^{4}\sum_{|m^j|=2:m^j \in \Gamma_j}^{N} w^{m^j}(t)e^{(m^j, \tau)} + \sum_{|m|=2, m \neq m^j, j=\overline{0,4}}^{N} w^m(t)e^{(m,\tau)}.$$

Then

$$\hat{G}(t, \tau) = w_0(t) + \sum_{i=1}^{4} w_i(t)e^{\tau_i} + \sum_{j=0}^{4}\sum_{|m^j|=2: m^j \in \Gamma_j}^{N} w^{m^j}(t)e^{\tau_j} + \sum_{|m|=2, m \neq m^j, j=\overline{0,4}}^{N} w^m(t)e^{(m,\tau)}.$$

Therefore, the embedding operation acts only on the resonant exponentials and replaces them with a unit or exponents $e^{\tau_j}$ of the first dimension according to the rule:

$$\left( e^{(m,\tau)}|_{m\in\Gamma_0} \right)^\wedge = e^0 = 1, \ \left( e^{(m,\tau)}|_{m\in\Gamma_j} \right)^\wedge = e^{\tau_j}, \ j = \overline{1,4}.$$

Therefore, the right-hand sides of iterative problems $(9_k)$ (if they solve sequentially) may not belong to the space $U$. Then, according to [6] (p. 234), the right-hand sides of these problems must be embedded in $U$ according to the above rule. As a result, we obtained the following problems:

$$Lz_0\,(t,\tau) \equiv \sum_{j=1}^{4} \lambda_j\,(t)\,\frac{\partial z_0}{\partial \tau_j} - A(t)z_0 - R_0 z_0 = h\,(t)\,,z_0\,(t_0,0) = z^0; \tag{$\overline{9}_0$}$$

$$Lz_1\,(t,\tau) = -\frac{\partial z_0}{\partial t} + \left[ \frac{g(t)}{2}\,(e^{\tau_2}\sigma_1 + e^{\tau_3}\sigma_2)\,z_0 \right]^\wedge + R_1 z_0, z_1\,(t_0,0) = 0; \tag{$\overline{9}_1$}$$

$$Lz_2\,(t,\tau) = -\frac{\partial z_1}{\partial t} + \left[ \frac{g(t)}{2}\,(e^{\tau_2}\sigma_1 + e^{\tau_3}\sigma_2)\,z_1 \right]^\wedge + R_1 z_1 + R_2 z_0, z_0\,(t_0,0) = 0; \tag{$\overline{9}_2$}$$

$$\cdots$$

$$Lz_k\,(t,\tau) = -\frac{\partial z_{k-1}}{\partial t} + \left[ \frac{g(t)}{2}\,(e^{\tau_2}\sigma_1 + e^{\tau_3}\sigma_2)\,z_{k-1} \right]^\wedge + R_k z_0 + \dots + R_1 z_{k-1},$$
$$z_k\,(t_0,0) = 0, k \geq 1 \tag{$\overline{9}_k$}$$

(images of linear operators $\frac{\partial}{\partial t}$ and $R_\nu$ do not need to be embedded in the space $U$, since these operators act from $U$ to $U$). Such a replacement will not affect the construction of an asymptotic solution to the original problem (1) (or its equivalent problem (2)), since the narrowing $\tau = \frac{\psi(t)}{\varepsilon}$ of the series of problems $(\overline{9}_k)$ will coincide with the series of problems $(9_k)$ (see [6], pp. 234–235).

It is easy to show that applying Theorem 1 to iterative problems $(\overline{9}_k)$, we can find their solutions uniquely in the space $U$. As a result, we can construct series (7) with coefficients $z_k(t,\tau) \in U$. As in [23] (pp. 303–308), we proved the following statement.

**Theorem 2.** *Suppose that conditions (1)–(2) are satisfied for the Equation (2). Then, when $\varepsilon \in (0,\varepsilon_0](\varepsilon_0 > 0$ is sufficiently small) the Equation (2) has a unique solution $z(t,\varepsilon) \in C^1([t_0,T],\mathbb{C})$; at the same time there is the estimate*

$$||z(t,\varepsilon) - z_{\varepsilon N}(t)||_{C[t_0,T]} \leq c_N \varepsilon^{N+1}, \ \forall N = 0,1,2,\ldots,$$

*where $z_{\varepsilon N}(t)$ is the narrowing (for $\tau = \frac{\psi(t)}{\varepsilon}$) $N$-th partial sum of the series (7) (with coefficients $z_k\,(t,\tau) \in U$ satisfying the iterative problems $(\overline{9}_k)$), and the constant $c_N > 0$ does not depend $\varepsilon$ on $\varepsilon \in (0,\varepsilon_0]$.*

## 5. Construction of the Solution of the First Iteration Problem in the Space $U$

Using Theorem 1, we will tried to find a solution to the first iterative problem $(\overline{9}_0)$. Since the right-hand side $h\,(t)$ of the equation $(\overline{9}_0)$ satisfies condition (11), this equation has (according to (15)) a solution in the space $U$ in the form

$$z_0\,(t,\tau) = z_0^{(0)}\,(t) + \alpha_1^{(0)}\,(t)\,e^{\tau_1}, \tag{17}$$

where $\alpha_1^{(0)}\,(t) \in C^\infty\,([t_0,T]\,,\mathbb{C})$ are arbitrary function, $z_0^{(0)}\,(t) = -\frac{h(t)}{\lambda_1(t)}$. Subordinating (17) to the initial condition $z_0\,(t_0,0) = z^0$, we have

$$z_0^{(0)}\,(t_0) + \alpha_1^{(0)}\,(t_0) = z^0 \ \Leftrightarrow \ \alpha_1^{(0)}\,(t_0) = z^0 + \lambda_1^{-1}\,(t_0)\,h\,(t_0)\,.$$

To fully calculate the function $\alpha_1^{(0)}(t)$, we pass to the next iterative problem $(\overline{9}_1)$. Substituting the solution (17) of the equation $(\overline{9}_0)$, into it, we arrived at the following equation:

$$
\mathbf{L}\,z_1(t,\tau) = -\frac{d}{dt}\left(z_0^{(0)}(t)\right) - \frac{d}{dt}\left(\alpha_1^{(0)}(t)\right)e^{\tau_1} + \frac{K(t,t)\,z_0^{(0)}(t)}{\lambda_4(t)}e^{\tau_4} -
$$

$$
-\frac{K(t,t_0)\,z_0^{(0)}(t_0)}{\lambda_4(t_0)} + \frac{g(t)}{2}\left(e^{\tau_2}\sigma_1 + e^{\tau_3}\sigma_2\right)\left(z_0^{(0)}(t) + \alpha_1^{(0)}(t)\,e^{\tau_1}\right) + \tag{18}
$$

$$
+\frac{K(t,t)\,\alpha_1^{(0)}(t)}{\lambda_1(t)}e^{\tau_1} - \frac{K(t,t_0)\,\alpha_j^{(0)}(t_0)}{\lambda_1(t_0)},
$$

(here we used the expression $(6_1)$ for $R_1 z(t,\tau)$ and took into account that when $z(t,\tau) = z_0(t,\tau)$ in the sum $(6_1)$ only terms with $e^{\tau_1}$ and remain $e^{\tau_4}$). Let us calculate

$$
M = \left[\frac{g(t)}{2}\left(e^{\tau_2}\sigma_1 + e^{\tau_3}\sigma_2\right)\left(z_0^{(0)}(t) + \alpha_1^{(0)}(t)\,e^{\tau_1}\right)\right]^{\wedge} =
$$

$$
= \frac{1}{2}g(t)\left[\sigma_1\alpha_1^{(0)}(t)\,e^{\tau_2+\tau_1} + \sigma_2\alpha_1^{(0)}(t)\,e^{\tau_3+\tau_1} + \sigma_1 z_0^{(0)}(t)\,e^{\tau_2} + \sigma_2 z_0^{(0)}(t)\,e^{\tau_3}\right]^{\wedge}.
$$

Let us analyze the exponents of the second dimension included here for their resonance:

$$
e^{\tau_2+\tau_1}\big|_{\tau=\psi(t)/\varepsilon} = e^{\frac{1}{\varepsilon}\int_{t_0}^{t}(-i\beta'(\theta)+a(\theta))d\theta}, \qquad e^{\tau_3+\tau_1}\big|_{\tau=\psi(t)/\varepsilon} = e^{\frac{1}{\varepsilon}\int_{t_0}^{t}(+i\beta'(\theta)+a(\theta))d\theta},
$$

$$
-i\beta' + a = \begin{bmatrix} 0, \\ a, \\ -i\beta', \\ +i\beta', \\ \mu, \end{bmatrix} \Leftrightarrow \varnothing; \qquad +i\beta' + a = \begin{bmatrix} 0, \\ a, \\ -i\beta', \\ +i\beta', \\ \mu, \end{bmatrix} \Leftrightarrow \varnothing.
$$

Thus, exponents $e^{\tau_2+\tau_1}$ ang $e^{\tau_3+\tau_1}$ are not resonant. Then, for solvability the Equation (18) it is necessary and sufficient that the condition

$$
-\frac{d}{dt}\left(\alpha_1^{(0)}(t)\right) + \frac{K(t,t)\,\alpha_1^{(0)}(t)}{\lambda_1(t)} = 0
$$

is satisfied. Attaching the initial condition $\alpha_1^{(0)}(t_0) = z^0 + \lambda_1^{-1}(t_0)\,h(t_0)$, to this equation, we found uniquely the function

$$
\alpha_1^{(0)}(t) = \alpha_1^{(0)}(t_0)\exp\left\{\int_{t_0}^{t}\frac{K(s,s)}{\lambda_1(s)}ds\right\},
$$

and therefore, we uniquely calculate the solution (17) of the problem $(\overline{9}_0)$ in the space $U$. In this case, the leading term of the asymptotics of the solution to the problem (2) has the form

$$
z_{\varepsilon 0}(t) = z_0^{(0)}(t) + \alpha_1^{(0)}(t_0)\exp\left\{\int_{t_0}^{t}\frac{K(s,s)}{\lambda_1(s)}ds\right\}e^{\frac{1}{\varepsilon}\int_{t_0}^{t}\lambda_1(\theta)d\theta},
$$

where $\alpha_1^{(0)}(t_0) = z^0 + A^{-1}(t_0)\,h(t_0)$, $z_0^{(0)}(t) = -\lambda_1^{-1}(t_0)\,h(t)$.

**Example.** *Consider a model problem*

$$
\varepsilon\frac{dz}{dt} = -z - \varepsilon\cos\frac{t^2+t}{\varepsilon}z - \int_{t_0}^{t}e^{\frac{-2(t-s)}{\varepsilon}}\cdot t\cdot s\cdot z(s,\varepsilon)ds + h(t),\ z(t_0,\varepsilon) = z^0,\ t\in[t_0,T](t_0\geq 0), \tag{19}
$$

*were $a(t) \equiv 1, \mu(t) \equiv -2, \beta(t) \equiv t^2 + t, K(t,s) \equiv t \cdot s$. The main term of the asymptotic solution of this problem has the form*

$$z_{\varepsilon 0}(t) = h(t) + [z^0 - h(t_0)] \exp\left[\frac{t_0^3 - t^3}{3}\right] exp\left[\frac{t - t_0}{\varepsilon}\right]. \tag{20}$$

For $\varepsilon \to +0$ the function $z_{\varepsilon 0}(t)$ tends to the solution of the degenerate equation $-\bar{z} + h(t) = 0$ uniformly on any interval $[t_0 + \delta, T](0 < \delta \leq T - t_0)$ and at the point $t = t_0$ takes on the value $z_{\varepsilon 0}(t_0) = z^0$. It is seen from (20) that the leading term of the asymptotics of the solution to problem (19) does not depend on $\cos\frac{t^2 + t}{\varepsilon}$ and spectral value $\mu(t) \equiv -2$, but depends on the kernel $K(t,s) \equiv t \cdot s$. Further calculations show that already the asymptotic solution $z_{\varepsilon 1}(t) = z_{\varepsilon 0}(t) + \varepsilon z_1\left(t, \frac{\psi(t)}{\varepsilon}\right)$ of the first order will depend on both $\mu(t) \equiv -2$, and the frequency $\beta'(t) = 2t + 1$ of the rapidly oscillating cosine.

## 6. Conclusions

The function $z_{\varepsilon 0}(t)$ shows that when passing from a differential equation of type (1) ($K(t,s) \equiv 0$) to an integro-differential one ($K(t,s) \neq 0$), the main term of the asymptotic is influenced by the kernel $K(t,s)$ of the integral operator. However, the main term of the asymptotics is not affected by the spectral values of the integral operator $\mu(t)$ and rapidly oscillating coefficients. Their effects are detected when constructing the next approximation $z_{\varepsilon 1}(t)$.

**Author Contributions:** All authors contributed equally to this work. All authors have read and agreed to the published version of the manuscript.

**Funding:** This work was supported by grant No. AP05133858 of the Ministry of Education and Science of the Republic of Kazakhstan.

**Conflicts of Interest:** The funders had no role in the design of the study; in the collection, analyses, or interpretation of data; in the writing of the manuscript, or in the decision to publish the results.

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
