# Peer review of "Regularization Method for Singularly Perturbed Integro-Differential Equations with Rapidly Oscillating Coefficients and Rapidly Changing Kernels"

_axioms, doi:10.3390/axioms9040131_

Round 1
Reviewer 1 Report
Definitely the paper is not acceptable for publication.
First, the object of investigation is a very special type of differential equation. The author did not point out the importance of this type of equations. They only wrote that they generalized a result from two books and a paper written in the previous senture. It means the object of investigation is not a actual oe and it could be interesting t a small number of readers.
Second the paper is not written carefully. For example, it is cited [26] which is not in the refences, it is cited Eq. (20) but there is not such kind of equation.
Also, the provided results in the paper are not formulated well. The authors set up two Theorems, there an unsucesfull trial for the proof of the first one and no proof of the second one. There is one example which is illustrating nothing in the paper. For,example, the main conditions are not checked or proved for the provided equation.
Author Response
- The article considers not a differential equation, but an integro-differential one. The class of such equations with rapidly oscillating coefficients has not previously been considered from the point of view of the regularization method. The article has a theoretical focus for the subsequent generalization of S.A. Lomov and is of interest to specialists in singular perturbations.
- The reference to the source [26] has been updated and replaced by [23].
- The remark about equation (20) is true. We have numbered it in the revised article.
- Example (19) illustrates the capabilities of the algorithm we have developed. All conditions of this algorithm in this example are checked directly
Reviewer 2 Report
The revised version can be published in Axioms.
Author Response
Thank you for your review, all comments have been corrected.
Reviewer 3 Report
The revised version has included my comments and suggestions.
Author Response

(The authors gave the same response as above.)

Round 2
Reviewer 1 Report
The paper still is not appropriate for publication. The authors did not make any changes in the revised version according to my main complains. For example, they wrote in their answer “The article has a theoretical focus” but they did not added any theoretical proofs. Only some calculations are provided.
- The range of the all functions being coefficients in the studied equations are not given in one and the same way- for example about the function a(t) - on line 31 it is R, on linen20 it is (0,infinity).
- English has to be polished. For example, what does “bay” mean on page 4; what does “we will tried” on page 9 mean? The sentence on lines 33-34 has no meaning.
- The notation N_z is not defined. It is written something on line 46, that it depends on z, but how does it depend? At the same time in (12) N_H is used? What is it?
- The authors did not change the proof of Theorem 1. The prove has to contain two par according to the claim Nothing about necessary and sufficient part is given.
- What does dots in (19) mean?
- About the example- the authors insist that “Example (19) illustrates the capabilities of the algorithm we have developed. All conditions of this algorithm in this example are checked directly” But they did not checked any conditions , they did not illustrate the application of the theoretical results of both theorems, they only did some calculations obtaining only the main term of the asymptotic solution and nothing else. By this way they did not illustrate the practical application of the provided without any proofs their algorithm.
I continue to think that almost no reader of this journal will use the provided algorithm because it is about a very specific type of equation and the algorithm is very difficult to be applied.
Author Response
- The article considers not a differential equation, but an integro-differential one. The class of such equations with rapidly oscillating coefficients has not previously been considered from the point of view of the regularization method. The article has a theoretical focus for the subsequent generalization of S.A. Lomov and is of interest to specialists in singular perturbations.
- 4 page, 6 line from the bottom, footnote to formula [26] replaced by [23].
- 10 page, 3 line from the bottom substituted the number (20) for the function.
- Example (19) illustrates the capabilities of the algorithm we have developed. All conditions of this algorithm in this example are checked directly
This manuscript is a resubmission of an earlier submission. The following is a list of the peer review reports and author responses from that submission.
Round 1
Reviewer 1 Report
The main goal of the reviewed work is to extend Lomov's regularization method to a new class os problems – IVP’s for integro-differential equations with rapidly oscillating coefficients. The authors investigate the influence of the integral term and construct the asymtotics using regularization approach. The mathematical material is presented in detail and consistently, analytical calculations are clearly explained.
I have a number of comments, mostly of an editorial nature.
- The description of the problem statement contains notation from other articles, not defined by the authors (matrix B (t) -?). For rapidly decreasig kernels /mu(t)< 0 (?).
- The frase “singularly perturbed equations are the object of study by a number of Russian researchers [9–25]” in the lines 28,29 with the following references is strange. The references are related to specific class of problems and it can be clearly stated/
- The work needs English editing. I just suggest one example : Lines 68 -72
However, cannot be considered completely regularized, since the integral term has not been regularized in it. There are many cases of misapplication of arcticles and there are many misprints.
But the paper is interesting, and it contains a new result. I would recommend to publish it after corrections.
Reviewer 2 Report
Работа содержит новые результаты, который дополняет теорию сингулярых возмущений, выполнена на высоком научном уровне и может быть опубликована в журнале Axioms.
The work contains new results, which complements the theory of singular perturbations, performed at a high scientific level and can be published in the journal Axioms.Reviewer 3 Report
The paper discusses regularization method for integro differential equation involving oscillating coefficients and changing kernels. The paper is designed in deep theoretical template. I do not have any comment about the scientific results as they might be correct, but the presentation and organization of the paper are not adequately prescribed. I have difficulty in understanding many steps and expressions. The English language needs extensive editing as well.Using WORD in writing made the presentation so unclear. There are neither examples nor ending conclusion which make the paper purely of theoretical nature. Regardless of the scientific results, I will not recommend this paper in this form.
Reviewer 4 Report
please see the attachment

Reviewer 5 Report
- Page 1 (Eq. (1)): why is just such a type of the equations studied? It looks very artificial.
2. Page 1 (line 36): what is B(t)?
3. Page 2 (line 38): the first part of this phrase is trivial.
4. Page 2 (line 39): it should be "are" not "is".
5. Page 3 (line 67): why "the vector function"?
6. Page 3 (line 71-72): are J_{tilde Z} and J the same?
7. Page 3 (line 72): what does it mean "a class M_{epsilon}". A class of what?
8. Page 4 (line 78): maybe in (5)?
9. Page 4 (lines 78-79): what does it mean "the degree N_{z} depends on the element z"? What element?
10. Page 4 (line 79): what is the space U?
11. Page 5 (line 102): what is e_{4}?
12. Page 13: Th. 2 should be proven.
13. Illustrative example should be presented.
14. English should be improved. 15. Conclusions section should be added.